# Central Regulation of Metabolism by Growth Hormone

**DOI:** 10.3390/cells10010129

**Published:** 2021-01-11

**Authors:** Jose Donato, Frederick Wasinski, Isadora C. Furigo, Martin Metzger, Renata Frazão

**Affiliations:** 1Departamento de Fisiologia e Biofisica, Instituto de Ciencias Biomedicas, Universidade de Sao Paulo, Sao Paulo 05508-000, Brazil; frednefro@gmail.com (F.W.); doraclivatti@gmail.com (I.C.F.); metzger@icb.usp.br (M.M.); 2Departamento de Anatomia, Instituto de Ciencias Biomedicas, Universidade de Sao Paulo, Sao Paulo 05508-900, Brazil; rfrazao@usp.br

**Keywords:** cytokines, energy balance, GH, glucose homeostasis, hypothalamus

## Abstract

Growth hormone (GH) is secreted by the pituitary gland, and in addition to its classical functions of regulating height, protein synthesis, tissue growth, and cell proliferation, GH exerts profound effects on metabolism. In this regard, GH stimulates lipolysis in white adipose tissue and antagonizes insulin’s effects on glycemic control. During the last decade, a wide distribution of GH-responsive neurons were identified in numerous brain areas, especially in hypothalamic nuclei, that control metabolism. The specific role of GH action in different neuronal populations is now starting to be uncovered, and so far, it indicates that the brain is an important target of GH for the regulation of food intake, energy expenditure, and glycemia and neuroendocrine changes, particularly in response to different forms of metabolic stress such as glucoprivation, food restriction, and physical exercise. The objective of the present review is to summarize the current knowledge about the potential role of GH action in the brain for the regulation of different metabolic aspects. The findings gathered here allow us to suggest that GH represents a hormonal factor that conveys homeostatic information to the brain to produce metabolic adjustments in order to promote energy homeostasis.

## 1. Introduction

Growth hormone (GH) is a single-chain 191 amino acid protein (its major isoform) which is mostly secreted by somatotropic cells located in the anterior pituitary gland. GH presents a pulsatile secretion pattern that is controlled by hypophysiotropic hypothalamic neurons (Figure 1). The classical neuroendocrine neurons that regulate the pulsatile secretion of GH express either somatostatin (SST) or GH-releasing hormone (GHRH) [1,2]. While SST-expressing neurons inhibit GH secretion, GHRH stimulates GH synthesis and release (Figure 1). Accordingly, defects in GHRH signaling causes dwarfism due to impaired GH secretion [3]. Ghrelin (a GH-releasing peptide) is also a powerful endogenous GH secretagogue [4,5,6,7]. Ghrelin activates growth hormone secretagogue receptor (GHS-R) in the hypothalamus and pituitary to induce GH secretion [8,9].

The half-life of GH in the blood is short, which is evident from its pulsatile secretion pattern, but GH’s pulsatile secretion pattern is the most important regulator of circulating insulin-like growth factor 1 (IGF-1), whose bioavailability is more stable [10] due to the fact that most of the circulating IGF-1 is associated with IGF-1 binding proteins. The liver is mainly responsible for maintaining circulating IGF-1 levels (Figure 1). Thus, the activation of GH receptor (GHR) in hepatocytes stimulates IGF-1 synthesis and secretion as well as increasing serum concentrations of this hormone [11,12]. The GH-IGF-1 axis is critically involved in the control of growth (Figure 1). However, genetic ablation of GHR from the liver does not compromise body growth as much as that caused by GHR deletion in the entire body, despite a drastic reduction in circulating IGF-1 levels [13]. Therefore, somatic growth is also directly regulated by GHR signaling in different tissues and local IGF-1 synthesis [11,12,14].

The strong relationship between circulating levels of GH and IGF-1 confounds the determination of the specific effects induced by each hormone individually, especially when considering the widespread expression of their receptors. However, with the generation of tissue-specific GHR- or IGF-1 receptor (IGF-1R)-deficient mice, the precise physiological role of each of these receptors in different tissues has been described [14,15,16]. In this regard, numerous studies produced tissue-specific knockout mice by deleting either GHR or IGF-1R from the bone, liver, adipose tissue, muscle, pancreatic β-cell and other organs [14,15,16]. Nevertheless, the distinct role of GHR signaling in the nervous system through the generation of brain-specific knockout mice has not been studied until recently [17].

In addition to its classical functions of regulating height, protein synthesis, tissue growth and cell proliferation, GH exerts profound effects on metabolism. Thus, GH stimulates lipolysis in white adipose tissue and antagonizes insulin effects on glycemic control [18,19,20,21]. Accordingly, GHR-deficient mice exhibit higher insulin sensitivity despite presenting increased percentages of body fat [13,15,16,21]. Since the classical target tissues of GH, including the liver, adipose tissue and muscle (Figure 1), are directly involved in the regulation of lipid and glucose metabolism, it is commonly assumed that the metabolic effects of GH are mediated by these organs. However, nowadays, it is well-established that the brain plays an important role regulating numerous metabolic aspects [22]. For example, target deletions of insulin or leptin receptors in the brain produce profound metabolic imbalances [23,24,25]. The brain is able to regulate systemic metabolism via the sympathetic and parasympathetic nervous systems. In addition, hypothalamic control of pituitary hormones, including those of the thyroid and adrenal axes, indirectly modulates whole-body metabolism [26,27]. In this present review, the current knowledge about the potential role of GH action in the brain for the regulation of different metabolic aspects is summarized, with a special focus on studies produced by our research group.

## 2. GH Action in the Brain

It has long been known that GHR is expressed in the brain [28,29]. Initially, GHR expression was described in hypothalamic areas that contain hypophysiotropic neurons that regulate pituitary GH secretion. In this context, previous studies have shown that a large percentage of SST neurons in the paraventricular (PVH) and periventricular (PV) hypothalamic nuclei express *Ghr* mRNA [28]. This finding is in accordance with the role of hypothalamic neurons regulating pituitary GH secretion via short negative feedback loops [1,30]. GHR is also amply expressed in the arcuate nucleus (ARH), which is the principal hypothalamic region that hosts GHRH-expressing neurons [28,31]. However, only a small percentage of GHRH neurons seem to contain GHR [32,33], whereas *Ghr* mRNA is abundantly detected in ARH neurons that express neuropeptide Y (NPY) [34,35]. Our research group has used the capacity of an acute GH injection to induce phosphorylation of the signal transducer and activator of transcription-5 (pSTAT5) as an alternative method to identify GH-responsive neurons [36,37]. Using this approach, the presence of GH-responsive neurons in the ARH, PVH and PV was confirmed. Furthermore, numerous additional brain areas that are directly responsive to GH were identified [36,37]. Thus, the widespread distribution of GH-responsive neurons in the brain suggests that a broad array of neural functions can be modulated by the direct action of GH [38] (Figure 2).

### Distribution of GH-Responsive Neurons in Mouse and Rat Brains

Systemically or centrally injected GH induces phosphorylation of the signal transducer and activator of transcription 5 (pSTAT5) in several brain areas of rats and mice [36,37] (Figure 2). GH-induced pSTAT5 was detected only in neurons [36], although it is possible that other cell types such as epithelial or glial cells are also responsive to GH. GH-induced pSTAT5 cells were found in the lateral septum, bed nucleus of the stria terminalis (BNST), paraventricular nucleus of the thalamus, amygdala (mostly in the medial and central subdivisions) and dorsal raphe nucleus, which are brain structures involved in the regulation of behaviors, emotions, limbic information and valence monitoring. In line with the anatomical distribution of GH-responsive cells, previous studies have described a ghrelin-GH axis in the amygdala that controls fear memory formation (Figure 2), possibly contributing with excessive fear memory typical of post-traumatic stress disorder [39,40].

GH administration also leads to several pSTAT5-positive cells in the hippocampus, a key brain structure for memory formation [36,37] (Figure 2). GH replacement improves memory function in adults with childhood-onset GH deficiency [41,42]. Accordingly, GH modulates excitatory synaptic transmission in the hippocampus [43,44,45]. Interestingly, brain-specific *Stat5a*/*Stat5b* knockout mice exhibit memory deficits [46], suggesting that STAT5 is a downstream signaling pathway potentially recruited by GHR to modulate memory. Paradoxically, global GHR knockout mice are protected from age-related decline in memory retention [47,48]. Additionally, GH overexpression leads to poor spatial learning and memory, whereas transgenic expression of a GHR antagonist caused improved learning in twelve-month-old male mice [49]. These apparent contradictory results can be explained by the fact that GHR signaling regulates insulin sensitivity and there is plenty of evidence indicating that brain insulin action plays a major role in regulating memory and is likely involved in the pathophysiology of Alzheimer’s disease [50,51,52]. Since global GHR knockout mice and GHR antagonist transgenic mice exhibit improved insulin sensitivity, whereas GH overexpression causes insulin resistance [21,53], the changes in memory performance in these mouse models are probably related to alterations in insulin action rather than a direct role of GHR signaling in the hippocampus.

Among all brain regions that express GHR, the hypothalamus displays the highest density of neurons responsive to GH [36,37]. Of note, GH-responsive neurons are particularly abundant in hypothalamic areas considered key players in the control of metabolism, including the ARH, PVH, ventromedial nucleus (VMH), dorsomedial nucleus (DMH) and lateral hypothalamic areas. Thus, GH action on these hypothalamic centers strongly indicates that the central regulation of energy and glucose homeostasis can be modulated by GHR signaling (Figure 2).

## 3. Central Regulation of Metabolism by GH

### 3.1. GH Regulates Food Intake

Findings from earlier studies have suggested that GH action in the brain has an orexigenic effect [54]. Accordingly, transgenic mice overexpressing GH in the central nervous system develop hyperphagia-induced obesity [55]. GH-transgenic carps also exhibit increased food intake [56]. Further evidence of the orexigenic effect of GH came from studies that performed intracerebroventricular injections of GH in wild-type mice and observed higher food intake compared to vehicle-injected animals [17,55]. It makes sense from an evolutionary point of view to associate growth with increased hunger; therefore, the energy required for GH-stimulated anabolic processes is guaranteed.

GH-induced increase in food intake seems to be mediated by ARH neurons that co-express NPY and agouti-related peptide (AgRP), which is a well-known neuronal population that stimulates feeding [57,58]. In accordance, GH overexpression in mice and fish induces upregulation of *Agrp* and/or *Npy* mRNA levels in the hypothalamus [55,56,59]. Furthermore, an acute GH injection increases *Agrp* and *Npy* mRNA levels in the hypothalamus of wild-type mice [17]. This is likely a direct effect of GH since approximately 95% of ARH AgRP/NPY neurons express *Ghr* mRNA or GH-induced pSTAT5 [17,34,35] (Figure 3A). Additionally, GH depolarizes the membrane potential of AgRP/NPY neurons, even in the presence of synaptic blockers [17]. In favor of the assumption that GH stimulates the activity of AgRP neurons in humans, circulating GH or IGF-1 levels exhibit a positive correlation with plasma AgRP levels in humans [60]. Moreover, plasma AgRP levels are high in patients with acromegaly, and pharmacological or surgical treatment reduces both GH secretion and plasma AgRP concentration [60].

Neurons that express proopiomelanocortin (POMC) in the ARH represent another key neuronal population involved in the regulation of feeding [61]. Approximately 60% of ARH POMC neurons exhibit pSTAT5 after an intracerebroventricular GH injection [62] (Figure 3A). Nevertheless, neither GH overexpression nor intracerebroventricular GH injection induce changes in hypothalamic *Pomc* expression [17,55,56]. Furthermore, GHR ablation in POMC-expressing cells does not elicit significant alterations in food intake in ad libitum fed mice nor after a period of food deprivation [62]. It is well documented that the injection of 2-deoxy-D-glucose (2DG) in rodents produces a glucoprivic condition which is followed by an acute increase in food intake [63,64,65]. Although 2DG injection increases the expression of *Npy* mRNA in the ARH, which could explain the 2DG-induced hyperphagia [65], another study suggested that NPY/AgRP neurons are not critical for the feeding responses to 2DG [64]. In this context, glucoprivic-induced hyperphagia was evaluated in knockout mice for GHR specifically in AgRP- or POMC-expressing cells. Notably, 2DG-induced hyperphagia is attenuated in mice lacking GHR either in AgRP neurons [17] or in POMC cells [62]. Thus, GH regulates glucoprivic-induced hyperphagia via these two major ARH neuronal populations.

Cholinergic neurons in the DMH and diagonal band of Broca (HBD) regulate food intake [66,67]. Choline acetyltransferase (ChAT) is a marker of cholinergic neurons and a population of ChAT-expressing neurons is also found in the ARH. ChAT neurons in the ARH co-express POMC and tyrosine hydroxylase (TH) [68,69]. Interestingly, 60% and 84% ChAT neurons in the ARH and DMH, respectively, exhibit GH-induced pSTAT5 [70] (Figure 3A). To determine whether GH action on cholinergic cells regulates food intake and metabolism, a ChAT-specific GHR knockout mouse was generated. Although these mutants exhibited reduced hypothalamic *Pomc* mRNA expression when exposed to a high-fat diet, ablation of GHR in cholinergic cells caused no metabolic consequences [70]. Interestingly, HBD ChAT neurons in mice are not responsive to GH, whereas 50% of cholinergic neurons in the rat HBD displayed GH-induced pSTAT5 [37]. Thus, it is possible that HBD ChAT-expressing neurons mediate the effects of GH on food intake in rats but not in mice.

Administration of the stomach-derived hormone ghrelin stimulates both GH secretion [4,5,6,7] and food intake [71]. However, the effects of ghrelin on food intake may be partially mediated by GH. Accordingly, GH action in the brain regulates the expression of GHS-R [72]. Noteworthily, GH-deficient or GHR-knockout mice manifest a blunted feeding response to ghrelin [71,73]. Pregnancy and lactation are physiological conditions that induce significant increases in food intake [74,75]. Remarkably, brain-specific GHR ablation decreases food intake and body adiposity during pregnancy, without affecting these parameters in non-pregnant or lactating mice [76]. Therefore, GH action in the brain regulates pregnancy-induced hyperphagia.

### 3.2. GH Action in the Brain Modulates Insulin Sensitivity and Glucose Homeostasis

Several hormones modulate whole-body glucose homeostasis through their action on the brain [77,78]. For example, leptin exerts its major effects on insulin sensitivity via the central nervous system [61]. Since GH also regulates insulin sensitivity, part of this effect may be mediated by the brain as well. Previous studies investigated the physiological role of GH signaling in leptin receptor (LepR)-expressing cells [79,80]. Notably, GHR ablation in LepR-expressing cells led to impaired hepatic insulin sensitivity [79,80]. In contrast, inactivation of GHR in specific populations of LepR-expressing neurons, including AgRP, POMC and steroidogenic factor 1 (SF1) cells, did not cause significant effects on glucose tolerance and insulin sensitivity [17,62,76,80]. Thus, a still-undisclosed population of LepR-expressing cells might mediate GH actions on insulin resistance.

Approximately one third of the cholinergic pre-ganglionic parasympathetic neurons located in the dorsal motor nucleus of the vagus (DMX) express pSTAT5 after an acute GH injection [70]. DMX neurons are able to regulate pancreatic hormone secretion as well as hepatic glycogenolysis and glucose production. However, GHR ablation in cholinergic neurons does not produce alterations in glucose homeostasis in mice consuming normal chow or high-fat diets [70]. Thus, further studies are needed to determine the physiological role of GHR signaling in this population of DMX neurons.

Pregnant animals usually develop a transient insulin resistance which is compensated by increases in glucose-stimulated insulin secretion [81]. Hormones secreted during pregnancy such as prolactin and placental lactogens play a prominent role in gestational metabolic adaptions, especially inducing the expansion of pancreatic beta-cells [82,83]. Since GH secretion in humans and rodents augments during pregnancy [84], GH action in the brain may also modulate glucose homeostasis in this condition. Remarkably, systemic insulin sensitivity is greatly improved in pregnant mice lacking GHR in the brain or in LepR-expressing cells [76]. Thus, while prolactin secretion promotes beta-cell adaptations during pregnancy, central GH action is critical to induce the typical insulin resistance observed in pregnant mice.

Together with glucocorticoids, noradrenaline and glucagon, GH is considered a counter-regulatory hormone that is secreted during hypoglycemia [77,85,86]. Accordingly, defects in GH secretion favor spontaneous hypoglycemia and impair the counter-regulatory response (CRR) [87,88,89]. Neurons in the VMH are responsive to GH [36]. Additionally, VMH contains glucose-sensing neurons that represent a key relay station in the neural circuitry that produces the CRR [77,78]. Using SF1 expression to drive VMH-specific deletion of GHR in mice, reduced glycemia was observed in mutant mice treated with insulin compared to control animals, without affecting insulin sensitivity. In addition, the CRR induced by 2DG is significantly attenuated in mice carrying ablation of GHR in VMH cells [80]. Inactivation of GHR in LepR-expressing cells also impairs the CRR to hypoglycemia [80], which is in line with the high degree of co-localization between SF1 and LepR expression in the VMH [90]. Collectively, these findings indicate that central GH action in SF1/LepR-positive neurons is relevant for recovery from hypoglycemia.

### 3.3. Central GH Action Regulates the Metabolic Responses to Calorie Restriction

#### 3.3.1. Central GHR Signaling Modulates Calorie Restriction-Induced Changes in Energy Expenditure

GH secretion increases during prolonged food restriction or fasting [85,89]. However, the precise role of GH action during these situations has not been completely understood until recently. It is well known that AgRP/NPY neurons become activated during calorie restriction and these neurons play an important role in suppressing energy expenditure and inducing hunger [57,58,91]. Remarkably, AgRP-specific GHR ablation attenuates fasting-induced activation of ARH AgRP/NPY neurons [17]. Mice carrying genetic ablation of GHR in AgRP neurons are unable to develop important metabolic responses to calorie restriction. In this regard, while control animals present activation of the adrenal axis and suppression of thyroid function, reproduction and thermogenesis during food restriction, the absence of GH action on AgRP cells impairs these neuroendocrine adaptations [17]. Consequently, compared to control animals, the energy expenditure of mice carrying AgRP-specific GHR ablation is less suppressed by calorie restriction, resulting in higher weight loss [17]. Thus, GH action on AgRP neurons represents a starvation signal that triggers energy-saving neuroendocrine adaptations to conserve body energy stores. Notably, administration of the GHR antagonist pegvisomant in C57BL/6 mice was able to reproduce the phenotype of the animals with AgRP-specific GHR ablation since it partially prevented the progressive reduction in energy expenditure during food restriction [17]. Ablation of the *Stat5a/b* genes in AgRP neurons also partially prevents the neuroendocrine adaptations induced by calorie restriction, indicating STAT5 transcription factors as the major downstream signaling pathway recruited by GHR to induce these effects [92]. It is worth mentioning that ablation of GHR in POMC- or SF1-expressing cells does not produce significant effects in the metabolic responses to calorie restriction [62,80], whereas GHR deletion in the entire brain or restricted to LepR-expressing cells reproduces the phenotype exhibited by AgRP-specific knockout mice during calorie restriction [17]. Therefore, the central metabolic responses induced by GH during calorie restriction seem to be solely mediated by AgRP neurons and the ablation of GHR in additional neuronal populations causes no further effects.

#### 3.3.2. Central GH Action Is Necessary to Maintain Blood Glucose Levels During Food Restriction

The maintenance of glycemia during prolonged food restriction requires numerous metabolic adaptations. The enzyme ghrelin O-acyltransferase (GOAT) is necessary for the generation of biologically active ghrelin. GOAT knockout mice present normal glycemia in ad libitum fed conditions, but after a few days of 60% calorie restriction, they exhibit hypoglycemia that can lead to death [89]. The lack of acyl-ghrelin (active form) in GOAT knockout mice prevents calorie-restriction-induced increases in plasma GH levels. Remarkably, GH replacement in GOAT knockout mice precludes the decrease in blood glucose levels during food restriction [89]. Other studies confirmed profound hypoglycemia in ghrelin-deficient mice during food restriction [93,94]. Thus, during prolonged food deprivation, ghrelin-induced GH secretion is necessary for the maintenance of glycemia and, consequently, to prevent death. Interestingly, blood glucose levels during 60% calorie restriction are not affected by GHR ablation in POMC or VMH neurons [62,80]. However, mice carrying inactivation of GHR in LepR- or AgRP-expressing neurons present a reduction in blood glucose levels during food restriction compared to control animals [17]. Therefore, GHR signaling in AgRP neurons that are also responsive to leptin is necessary for the maintenance of glycemia during prolonged food restriction.

### 3.4. Adaptation Capacity to Aerobic Exercise Is Affected by Central GHR Signaling

The hypothalamus not only regulates metabolism in basal conditions but also during exercise training. There is accumulating evidence indicating that VMH neurons are major regulators of metabolism during exercise. Hypothalamic ablation of SF1 prevents the beneficial metabolic effects of exercise [95]. Inactivation of *Socs3* gene from SF1/VMH neurons also impairs exercise performance [96]. Of note, the suppressor of cytokine signaling 3 (SOCS3) is a major negative regulator of cytokine signaling [97,98]. Thus, SOCS3 ablation likely affects the responses to different hormones in the VMH, including leptin and GH. To investigate whether GH signaling in the hypothalamus or specifically in VMH neurons alters acute and chronic metabolic adaptations to exercise, mice carrying ablation of GHR in LepR or SF1 cells were subjected to 8 weeks of treadmill running training [99]. Remarkably, while GHR deletion in LepR-expressing cells led to improved aerobic performance, GHR ablation in SF1 cells prevented improvements in running capacity [99]. These effects are possibly associated with modified glycemic responses to exercise, since SF1 GHR and LepR GHR knockout mice exhibited distinct changes in blood glucose levels in response to acute exercise [99].

## 4. Neurotropic Effects of GH on ARH Neurons

GH is an important growth factor in several peripheral tissues and also in the brain. As previously mentioned, GH not only modulates synaptic plasticity in the hippocampus [43,44,45] and amygdala [39], but this hormone also has important neurotropic effects on the development of ARH neurons that regulate metabolism [100,101]. It is well documented that leptin is required for the development of axonal projections of ARH neurons to post-synaptic targets such as the PVH [102,103]. Interestingly, GH- or GHR-deficient mice exhibit reduced axonal projections from ARH AgRP and POMC neurons to the PVH, similarly to that observed in leptin- or LepR-deficient mice [100,101]. However, GH or GHR deficiency leads to severe growth deficits as well as numerous metabolic and endocrine alterations. Thus, the study of mice carrying ablation of GHR specifically in LepR, AgRP or POMC neurons is an interesting approach to clarify whether GHR signaling, without those confounding factors, is required for the development of AgRP and POMC axonal projections [62,100]. Confirming a direct neurotropic effect of GH in ARH neurons, GHR ablation in LepR-expressing cells decreased the density of both AgRP and POMC axonal projections to hypothalamic post-synaptic targets. When GHR deletion was restricted to AgRP neurons, only AgRP axonal projections were reduced, whereas a normal POMC innervation was observed [100]. Of note, mice carrying ablation of *Stat5a/Stat5b* genes in AgRP neurons show normal AgRP axonal projections, suggesting that STAT5-independent signaling pathways are involved in the neurotropic effects of GH [92]. Furthermore, GHR ablation specifically in POMC neurons did not affect POMC axonal projections [62]. The percentage of ARH AgRP neurons that are responsive to GH is much higher than that of POMC neurons [17,62], which could explain the minor neurotropic effects of GH when the entire population of POMC neurons is analyzed. Therefore, the central regulation of metabolism by GH may involve the neurotropic effects of GH in hypothalamic neurons that control energy and glucose homeostasis. Whether these neurotropic effects are ubiquitously observed in other neuronal populations responsive to GH is still unknown.

## 5. Future Perspectives

Table 1 summarizes the published articles so far that used mouse models with ablation of GHR in specific neuronal populations to investigate the physiological role of central GH action. The importance of GHR signaling in several newly described populations of GH-responsive neurons remains undetermined [36,37]. Recently, our research group identified the neurochemical phenotype of GH-responsive cells in mouse PVH [104]. The PVH contains several neurochemically defined neuronal populations [105], so the identification of those that are responsive to GH allows understanding the possible physiological importance of GHR signaling in this hypothalamic nucleus. We found that 38%, 55%, 35% and 63% of TH, SST, thyrotropin-releasing hormone (TRH) and corticotropin-releasing hormone (CRH) neurons exhibited GH-induced pSTAT5, respectively [104] (Figure 3B). The majority of neuroendocrine SST, TRH and CRH neurons were responsive to GH, indicating that central GH signaling probably regulates somatotropic, thyroid and adrenal endocrine axes (Figure 2). However, non-neuroendocrine neurons were also responsive to GH in the PVH, including 67%, 32% and 74% of non-neuroendocrine TH, TRH and CRH PVH neurons, respectively [104]. This study gives an idea about the diversity of functions that central GHR signaling may regulate in one specific hypothalamic nucleus. However, follow-up physiological studies investigating the role of GHR signaling in each PVH neuronal population are warranted.

In another study, our research group disclosed that several neuronal populations that express TH are responsive to GH, including neurons in the ARH, PVH (Figure 3), PV and locus coeruleus (LC) [33]. Of note, LC is the major source of noradrenergic projections to the forebrain [107], which are vital for the control of physiological responses to stress. Importantly, GH-responsive neurons are also enriched in several other brain structures involved in stress responses, including BNST, the central nucleus of the amygdala and CRH-expressing cells in the PVH (Figure 3B). Thus, central GHR signaling likely plays a significant role in modulating the central responses to stress, including situations such as glucoprivation, food restriction and physical exercise. Therefore, based on the aforementioned neuroanatomical evidence, it is imperative that future studies investigate how central GH actions can influence the neurocircuits that trigger stress responses, assessing possible consequences on metabolism and behavioral regulation.

## 6. Conclusions

This review article presents an overview of recent research data indicating that central GHR signaling regulates metabolism, particularly in situations of metabolic stress. The high responsiveness to GH in hypothalamic and extra-hypothalamic neuronal populations that control metabolism, especially the ARH, VMH, and PVH, suggests that circulating GH levels may represent a cue that conveys homeostatic information to the brain to produce metabolic adjustments in order to promote energy homeostasis. Accordingly, GH secretion is elevated in numerous situations of metabolic stress, including hypoglycemia [85], prolonged calorie restriction [89], and exercise [108]. Elevated GH secretion is also observed in physiological situations characterized by additional metabolic exigencies such as puberty [106,109] and pregnancy [84]. Thus, in addition to its well-known effects of stimulating protein synthesis, tissue growth, and cell proliferation, GH should be considered a metabolic hormone which, besides acting in peripheral tissues, also fulfills important functions in the central nervous system. Finally, the recognition of the central effects of GH regulating metabolism can provide significant new contributions to the understanding of metabolic changes that occur in different physiological or pathological conditions.

## Figures and Tables

**Figure 1 cells-10-00129-f001:**
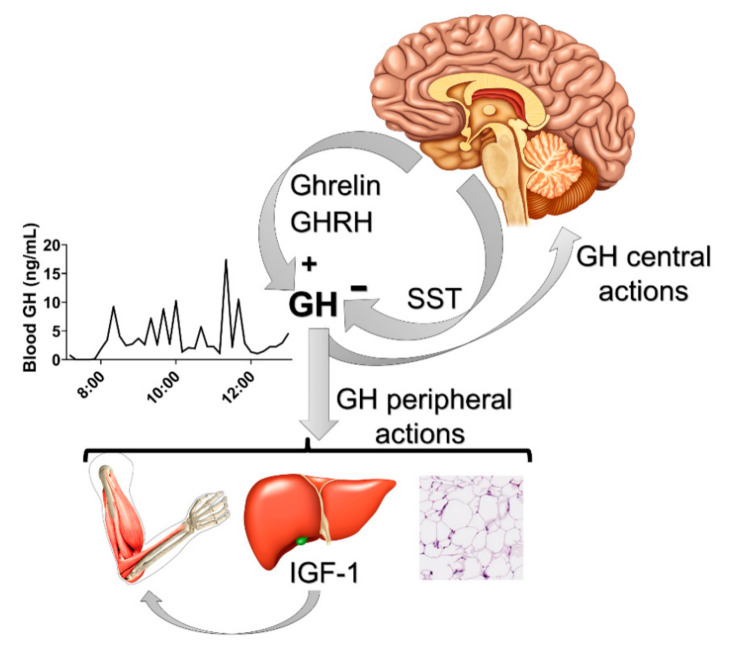
Neuroendocrine factors that control the pulsatile secretion of growth hormone (GH). Ghrelin and GH-releasing hormone (GHRH) stimulate pituitary GH secretion, whereas somatostatin (SST) inhibits it. The example of the pulsatile secretion of GH was obtained from a C57BL/6 eight-week-old male mouse after 36 serial blood collections in a 10-min interval (lights on at 7 am; 12-h light/dark cycle). Note that GH has central and peripheral actions (e.g., on the liver, muscle and white adipose tissue), in which the stimulation of insulin-like growth factor 1 (IGF-1) secretion from the liver plays a major role controlling somatic growth.

**Figure 2 cells-10-00129-f002:**
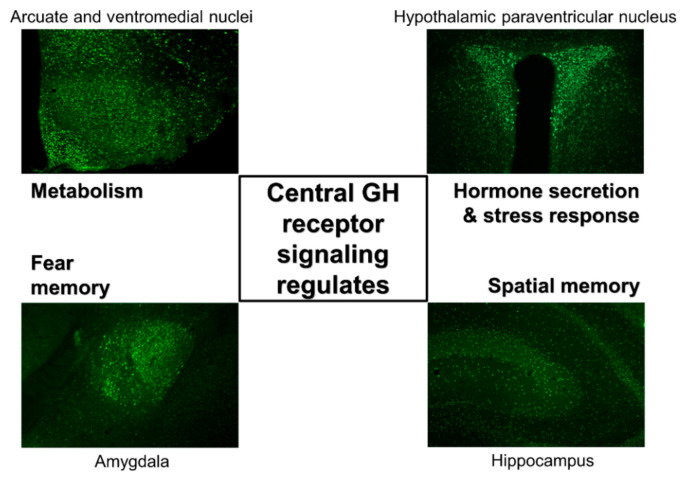
GH-responsive neurons are found in several brain structures, and GH receptor signaling regulates different neurological aspects. Coronal photomicrographs of different parts of the mouse brain showing immunoreactivity against the phosphorylation of the signal transducer and activator of transcription 5 after an acute GH injection. Previous studies indicated that GH receptor signaling in different brain areas regulates distinct physiological parameters. Although GH-responsive neurons are abundantly found in the paraventricular nucleus of the hypothalamus, the exact role played by GH in these cells is still unknown.

**Figure 3 cells-10-00129-f003:**
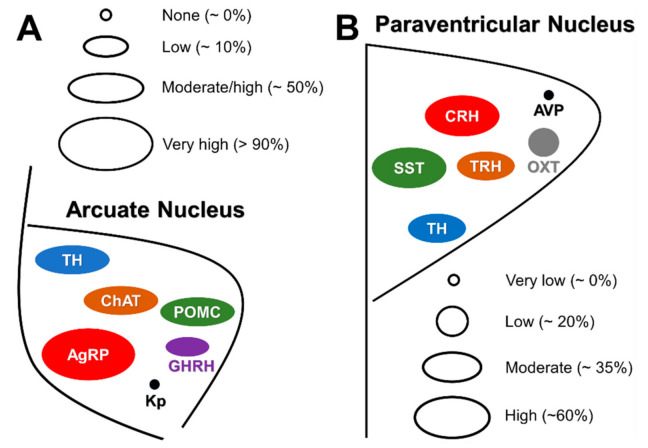
Neurochemical phenotype of GH-responsive neurons in the hypothalamic arcuate and paraventricular nuclei. (**A**) Scheme illustrating the approximate percentage of responsiveness to GH in arcuate nucleus neurons that express agouti-related peptide (AgRP), proopiomelanocortin (POMC), choline acetyltransferase (ChAT), tyrosine hydroxylase (TH), GH-releasing hormone (GHRH) and kisspeptins (Kp). (**B**) Scheme illustrating the approximate percentage of responsiveness to GH in hypothalamic paraventricular nucleus neurons that express somatostatin (SST), TH, corticotropin-releasing hormone (CRH), thyrotropin-releasing hormone (TRH), oxytocin (OXT) and vasopressin (AVP). The relative size of circles and ellipsoids indicates the approximate percentage of neurons in these nuclei which are responsive to GH.

**Table 1 cells-10-00129-t001:** Summary of the published articles that used mouse models with ablation of GH receptor in specific neuronal populations to investigate the physiological role of central GH action.

Neuronal Population	Physiological Aspects Regulated by Central GH Receptor Signaling	Reference
Nestin-derived cells (entire brain)	Neuroendocrine adaptations that affect energy expenditure during food restriction	[17]
	Regulation of GH secretion via a negative feedback loop	[33]
	Food intake, fat retention, as well as insulin and leptin sensitivity during pregnancy	[76]
LepR-expressing cells	Hepatic glucose production and insulin sensitivity	[79]
	Neuroendocrine adaptations that affect energy expenditure during food restriction	[17]
	Maintenance of glycemia during prolonged food restriction	[17]
	Glucoprivic hyperphagia	[17]
	Food intake, fat retention, as well as insulin and leptin sensitivity during pregnancy	[76]
	Aerobic performance and metabolic adaptations to chronic exercise	[99]
	Recovery from hypoglycemia and counter-regulatory response	[80]
	Trophic effects on the formation of POMC and AgRP axonal projections	[100]
TH-expressing cells	Regulation of GH secretion via a negative feedback loop	[33]
Dopamine transporter-expressing cells	No function identified yet	[33]
Dopamine β-hydroxylase-expressing cells	No function identified yet	[33]
AgRP-expressing neurons	Neuroendocrine adaptations that affect energy expenditure during food restriction	[17]
	Maintenance of glycemia during prolonged food restriction	[17]
	Glucoprivic hyperphagia	[17]
	Trophic effects on the formation of AgRP axonal projections	[100]
Cholinergic cells	No function identified yet	[70]
Kisspeptin-expressing neurons	Regulation of the hypothalamic expression of transcripts that modulate the hypothalamic-pituitary-gonadal axis	[106]
POMC-expressing neurons	Glucoprivic hyperphagia	[62]
SF1-expressing cells (VMH neurons)	Recovery from hypoglycemia and counter-regulatory response	[80]
	Aerobic performance and metabolic adaptations to exercise	[99]

## Data Availability

No new data were created or analyzed in this study. Data sharing is not applicable to this article.

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
