# Peer review of "Central Regulation of Metabolism by Growth Hormone"

_cells, 2021, doi:10.3390/cells10010129_

Round 1
Reviewer 1 Report
The review by Jose Donato Jr and his lab provides an excellent summary of work that has been conducted using mouse lines with alterations to the GH/IGF-1 axis (primarily with disruptions to the GHR in various brain cell types) in understanding the central regulation of metabolism by GH. While previous reviews exist summarizing mouse lines lacking GHR in non-neural tissues, there are no reviews that summarize the neural specific GHR knockout lines, thus this review fills an much needed space. On top of this, many of these lines and studies have been generated and conducted by the Donato lab, thus, this lab is supremely qualified to write this review.
The review is well written and organized I have only minor comments to consider.
1) There are many occasions where the authors use the wording "we" which is generally frowed about in reviewing the literiture. However, in this instance since the laboratory who wrote this review has completed much of the research summarized in the paper, I feel that in this case it is acceptable.
2) line 35, I do not believe that Ghrelin is defined correctly (Growth Hormone-RELeasINg peptide)... according to the orignial Nature paper where ghelin was discoverd and named, it was described as the following: [We designate the GH releasing peptide "ghrelin" (ghre is the proto-indo-european root of the word grow)]. However, i may be incorrectly enterpreting this passage?
3) lines 38 & 40, it would be helpful to the reader if the authors provide the half life of GH and IGF-1 with references.
4) figure 1, I assume the graph showing and example of "pulsitile" GH secretion is simply a cartoon example to demonstrate pulsitility and does not reflect and actual reading. If so it may be better to re-draw the graph to mimic actual measures (ie. the largest peak of GH is realeased at ~3am, not at ~11pm) just in case some readers try to enterpret more meaing into this graph.
5) line 76 replace "the" with "this"
6) lines 115-116, while several studies suggest that GH improves memory function, paradoxically, there are other studies that also suggest lack of GH action also improves memory. For example GHR-/- mice have imporved memory retention vs WT in inibitory avoidance tests (Kinney BA 2001 Physiol Behav 72:653-660) as well as water maze tests (Kinney-Forshee Ba 2004 Physiol Behav 80:589-94). Please reconsile if possible?
Author Response
1) There are many occasions where the authors use the wording "we" which is generally frowed about in reviewing the literiture. However, in this instance since the laboratory who wrote this review has completed much of the research summarized in the paper, I feel that in this case it is acceptable.
RESPONSE: We thank the reviewer for the constructive comments that improved the manuscript. We revised the entire manuscript to minimize the use of “we” in the text.
2) line 35, I do not believe that Ghrelin is defined correctly (Growth Hormone-RELeasINg peptide)... according to the orignial Nature paper where ghelin was discoverd and named, it was described as the following: [We designate the GH releasing peptide "ghrelin" (ghre is the proto-indo-european root of the word grow)]. However, i may be incorrectly enterpreting this passage?
RESPONSE: The reviewer is correct. We revised the manuscript and now we just say that ghrelin is a GH-releasing peptide.
3) lines 38 & 40, it would be helpful to the reader if the authors provide the half life of GH and IGF-1 with references.
RESPONSE: We added more information in this phrase and included a reference, as requested by the reviewer.
4) figure 1, I assume the graph showing and example of "pulsitile" GH secretion is simply a cartoon example to demonstrate pulsitility and does not reflect and actual reading. If so it may be better to re-draw the graph to mimic actual measures (ie. the largest peak of GH is realeased at ~3am, not at ~11pm) just in case some readers try to enterpret more meaing into this graph.
RESPONSE: The graph in figure 1 is indeed a real example of the pulsatile GH secretion in mice, which have a circadian rhythm opposite to that of humans (at least in relation to the pattern of activity, sleep and, consequently, GH secretion). In the figure legend we mention: “The example of the pulsatile secretion of GH was obtained from a C57BL/6 eight-week old male mouse after 36 serial blood collections in a 10-minutes interval (lights on at 7 am; 12-h light/dark cycle)”.
5) line 76 replace "the" with "this"
RESPONSE: The word was replaced.
6) lines 115-116, while several studies suggest that GH improves memory function, paradoxically, there are other studies that also suggest lack of GH action also improves memory. For example GHR-/- mice have imporved memory retention vs WT in inibitory avoidance tests (Kinney BA 2001 Physiol Behav 72:653-660) as well as water maze tests (Kinney-Forshee Ba 2004 Physiol Behav 80:589-94). Please reconsile if possible?
RESPONSE: As suggested by the reviewer, we added a paragraph commenting and discussing the studies that showed improved memory function in global GHR knockout mice: “Paradoxically, global GHR knockout mice are protected from age-related decline in memory retention [43,44]. Additionally, GH overexpression leads to poor spatial learning and memory, whereas transgenic expression of a GHR antagonist causes improved learning in twelve-month old male mice [45]. These apparent contradictory results can be explained by the fact that GHR signaling regulates insulin sensitivity and there is plenty of evidence indicating that brain insulin action plays a major role regulating memory and is likely involved in the pathophysiology of Alzheimer's disease [46-48]. Since global GHR knockout mice and GHR antagonist transgenic mice exhibit improved insulin sensitivity, whereas GH overexpression causes insulin resistance [17,49], the changes in memory performance in these mouse models are probably related to alterations in insulin action, rather than a direct role of GHR signaling in the hippocampus.”
Reviewer 2 Report
This is a very well composed review describing the central actions of GH on energy balance and glucose homeostasis.
This is a very well composed review describing the central actions of GH on energy balance, glucose homeostasis and insulin sensitivity.
The manuscript covers most of the studies in the area and is concise and easy to read. Moreover, the figures are very informative.
I only have minor comments.
In the abstract I miss a mention to the action of GH on food intake especially when there is a whole section on it in the main text.
In the same way, something similar happens to me with the action of GH over energy expenditure, it is commented on in the abstract but there is no section reefing to that in the main text. I suggest subdividing the section 3.3 into energy expenditure and another on hypoglycemia maintenance.
Line 273 Inactivation of Socs3 gene from SF1 cells, including VMH neurons. SF-1 neurons are only expressed in VMH.
As the authors say, GH is a counterregulatory hormone released during hypoglycemia. In this sense could be of interest that the authors consider to mention the work of Lee J over the role of central insulin on GH et al (JCI Insight. 2020 Aug 20;5(16):e135412).
Although it is well-known that several factors are able to regulate GH secretion, neurons that express either somatostatin (SST) or GH-releasing hormone (GHRH) are considered the most relevant regulators of the pulsatile secretion of GH [1,2]. In this sentence the authors propose that GHRH and SS are the main regulators of GH, however it has been demonstrated that ghrelin works as a GH secretagogue as potent or even more powerful than GHRH in laboratory rats (PMID: 11078999). The role of ghrelin in GH secretion in this sentence appears to be secondary to the canonical regulators of GH but ghrelin is also fundamental for its regulation, as important as GHRH.
The authors have done a good job with the references, however, those that refer to the secretion of GH by ghrelin are not very precise. An example of seminal works about this issue are: PMID: 11124868, PMID: 11089570, PMID: 11078999 and Kojima M et al 1999.
Moreover, in Line 130: The authors should consider add this reference that demonstrates an upregulation of AgRP and food intake in GH transgenic coho salmon (PMID: 26123591).
Author Response
In the abstract I miss a mention to the action of GH on food intake especially when there is a whole section on it in the main text.
RESPONSE: We thank the reviewer for his/her suggestions. In the abstract, we included the food intake as one of the metabolic aspects regulated by GH.
In the same way, something similar happens to me with the action of GH over energy expenditure, it is commented on in the abstract but there is no section reefing to that in the main text. I suggest subdividing the section 3.3 into energy expenditure and another on hypoglycemia maintenance.
RESPONSE: As suggested by the reviewer, we subdivided Section 3.3 in:
3.3.1. Central GHR Signaling Modulates Calorie Restriction-Induced Changes in Energy Expenditure
3.3.2. Central GH Action Is Necessary to Maintain Blood Glucose Levels During Food Restriction
Line 273 Inactivation of Socs3 gene from SF1 cells, including VMH neurons. SF-1 neurons are only expressed in VMH.
RESPONSE: We changed the sentence to avoid redundancy.
As the authors say, GH is a counterregulatory hormone released during hypoglycemia. In this sense could be of interest that the authors consider to mention the work of Lee J over the role of central insulin on GH et al (JCI Insight. 2020 Aug 20;5(16):e135412).
RESPONSE: We included the citation of the work of Lee et al.
Although it is well-known that several factors are able to regulate GH secretion, neurons that express either somatostatin (SST) or GH-releasing hormone (GHRH) are considered the most relevant regulators of the pulsatile secretion of GH [1,2]. In this sentence the authors propose that GHRH and SS are the main regulators of GH, however it has been demonstrated that ghrelin works as a GH secretagogue as potent or even more powerful than GHRH in laboratory rats (PMID: 11078999). The role of ghrelin in GH secretion in this sentence appears to be secondary to the canonical regulators of GH but ghrelin is also fundamental for its regulation, as important as GHRH.
RESPONSE: We rephrased the sentence to not erroneously suggest that GHRH is more potent than ghrelin to induce GH secretion: “The classical neuroendocrine neurons that regulate the pulsatile secretion of GH express either somatostatin (SST) or GH-releasing hormone (GHRH) [1,2]. While SST-expressing neurons inhibit GH secretion, GHRH stimulates GH synthesis and release (Figure 1). Accordingly, defects in GHRH signaling causes dwarfism due to impaired GH secretion [3]. Ghrelin (GH-releasing peptide) is also a powerful endogenous GH secretagogue [4]. Ghrelin activates growth hormone secretagogue receptor (GHS-R) in the hypothalamus and pituitary to induce GH secretion [5,6].”
The authors have done a good job with the references, however, those that refer to the secretion of GH by ghrelin are not very precise. An example of seminal works about this issue are: PMID: 11124868, PMID: 11089570, PMID: 11078999 and Kojima M et al 1999.
RESPONSE: These citations were included in the first paragraph of the manuscript and in the last paragraph of section 3.1.
Moreover, in Line 130: The authors should consider add this reference that demonstrates an upregulation of AgRP and food intake in GH transgenic coho salmon (PMID: 26123591).
RESPONSE: Thank you for the suggestion. The reference was included in the following paragraph: “GH-induced increase in food intake seems to be mediated by ARH neurons that coexpress NPY and agouti-related peptide (AgRP), which are a well-known neuronal population that stimulates feeding (Aponte et al., Nat Neurosci 14:351-355, 2011; Krashes et al., J Clin Invest 121:1424-1428, 2011). In accordance, GH overexpression in mice and fish induces upregulation of Agrp and/or Npy mRNA levels in the hypothalamus (Bohlooly et al., Diabetes 54:51-62, 2005; Zhong et al., Gen Comp Endocrinol 192:81-88, 2013; Kim et al., Mol Cell Endocrinol 413:178-188, 2015)”.